# Communicating risk in human-wildlife interactions: How stories and images move minds

**Sara K. Guenther**[ID][☯]*, **Elizabeth A. Shanahan**[☯]

Department of Political Science, Montana State University, Bozeman, MT, United States of America

☯ These authors contributed equally to this work.
* sara.guenther@montana.edu

## Abstract

Effectively communicating risk is critical to reducing conflict in human-wildlife interactions. Using a survey experiment fielded in the midst of contentious public debate over flying fox management in urban and suburban areas of Australia, we find that stories with characters (i.e., narratives) are more effective than descriptive information at mobilizing support for different forms of bat management, including legal protection, relocation, and habitat restoration. We use conditional process analysis to show that narratives, particularly with accompanying images, are effective because they cause emotional reactions that influence risk perception, which in turn drives public opinion about strategies for risk mitigation. We find that prior attitudes towards bats matter in how narrative messages are received, in particular in how strongly they generate shifts in affective response, risk perception, and public opinion. Our results suggest that those with warm prior attitudes towards bats report greater support for bat dispersal when they perceive impacts from bats to be more likely, while those with cool priors report greater support for bat protection when they perceive impacts from bats to be more positive, revealing 1) potential opportunities for targeted messaging to boost public buy-in of proposals to manage risks associated with human-wildlife interactions, and 2) potential vulnerabilities to disinformation regarding risk.

## Introduction

The present fallout from the global COVID-19 pandemic is a poignant reminder of how serious the consequences of human-wildlife interaction can be. Most emerging infectious diseases, such as the novel coronavirus, are the result of transmission of a pathogen from animals to humans that can result in human illness and death [1, 2]; other negative impacts of wildlife on humans range from destruction of property to general nuisance [3, 4]. Degradation of natural habitat and displacement of wildlife toward more urban areas not only increases the frequency of human-wildlife interactions, but also poses threats to wildlife health [5, 6]. These risks exacerbate human-wildlife conflict (HWC) and easily permeate already contentious political arenas and polarized debates among those seeking to prioritize the protection of wildlife and those

**Data Availability Statement:** The data underlying the results presented in the study have been deposited to Dryad and may be accessed at https://doi.org/10.5061/dryad.v6wwpzgtr.

**Funding:** EAS awarded and SKG funded through National Science Foundation CNH-L grant number

1716698, https://www.nsf.gov/. EAS and SKG both
awarded National Science Foundation RAPID grant
number 1914601, https://www.nsf.gov/ The
funders had no role in study design, data collection
and analysis, decision to publish, or preparation of
the manuscript.

**Competing interests:** The authors have declared
that no competing interests exist.

seeking to prioritize the protection of humans and property [7]. Therefore, solutions to HWCs
not only require scientific insights from biology, ecology and epidemiology, but also from the
social sciences to best understand drivers of policy support aimed to reduce both risk and con-
flict [8–11]. In this study, we develop an inclusive, social science model by incorporating four
theoretically-based concepts in the realm of risk communication: risk perception, affect heu-
ristic, narratives, and prior beliefs.

Many studies of HWC policy and management are premised on a model of risk, whereby
support for policies is mediated through the perception of wildlife risk to humans [12].
Because the long-standing definition of risk entails measures of adverse effects (severity of
impact x likelihood of impact; the dread and uncertainty factors) [13, 14], the majority of
HWC studies assess risk solely on negative anthropocentric outcomes [15]. However, wildlife
provide a range of benefits in the form of ecosystem services and, in some cases, contributions
to human health and well-being [16]. More recently, the concept of coexistence has broadened
HWCs (now called HWCCs) to include the dual role of risk with wildlife: a purveyor of harm
to humans and a positive source of benefits for the environment [16, 17]. While there have
been advances in conceptualizing the interplay between negative harm and positive benefits
through the coexistence scholarship, there remains a need for more precise modeling that
accounts for the polarity inherent in risk perception for HWCCs [18].

When judging the risks and benefits of interacting with nature, including wildlife, people
rely on an intuitive affective assessment of the risk [19]. This 'affect heuristic' has been found
to be influential in the cognitive process of risk perception [20, 21], with the strength of the
positive-negative valence of affect largely predicting the intensity of positive benefits and nega-
tive impacts of risk [22]. Yet, the typical emotions accounted for in studies of risk perception
are negative, such as afraid, worried, concerned, and angry [23, 24]. Additionally, the import
of affect in communication rests on the concept of transporting the audience through an affec-
tive experience [25]. Indeed, modeling both positive and negative affect, as well as perceived
benefits and risks, has deepened our understanding of information-seeking behavior in other
areas of environmental communication [26]. Therefore, to advance a more complete represen-
tation of the HWCC system, the range of positive to negative affect needs to be recognized and
operationalized in tandem with the polarity of benefits and harm in risk perception. Public
opinion about HWCC management may be more clearly understood through this valence
approach to the critical mediating factors, affect and risk perception.

As critical as affect and risk perception are in understanding policy support, they are not
stable or static concepts. Two forces carry the potential to profoundly shape these concepts: (i)
risk communication messaging [27] and (ii) prior attitudes and experiences (e.g., risk experi-
ence) [28, 29]. Regarding the former, HWCC management often includes educational messag-
ing [30] to encourage certain behaviors. Additionally, a vibrant area in risk communication
studies not only finds that narrative-based risk communication is generally persuasive [31],
but that stories influence affective responses and risk perceptions across multiple risk domains:
natural hazards [32], health [33], climate change [34], and HWCC [35, 36]. Furthermore,
explorations of risk communication in HWCC largely rely on printed communication materi-
als such as brochures, newsletters, and media accounts [37]. However, people are increasingly
using social media as a source of information [38], which includes affective portrayal through
emojis [39] and images [40]. A few prior studies suggest that the way wildlife is portrayed in
images influences support for wildlife conservation, without exploring the mediating mecha-
nisms involved [37, 41]. Advancing the efficacy of risk communication efforts for HWCCs
and other risk domains includes testing the power of educational information compared to
narrative-based communication, testing the power of images used in a simulated social media
venue, and examining the role of mediating and moderating variables.

Priors are the values, knowledge, and experiences that serve as drivers in affective responses and evaluations of information and risk. Prior attitudes determine how risk messages are received and interpreted [42–44]. Risk messages may affirm prior strong beliefs, challenge prior strong beliefs, or persuade an audience with neutral priors to adopt a particular belief. Scholars have approached conceptualizing priors through studies of confirmation/disconfirmation bias [45, 46], congruence/incongruence between narratives and belief systems [46], selective exposure [47, 48], and identity-protection cognition [49]. HWCC studies primarily rely on prior experiences as the proxy for priors writ large. While important covariates, experiences with and exposure to wildlife is insufficient as a measure for priors. Given the valanced nature of affect and risk perception, a valanced measure of priors would logically be most appropriate. The American Election Studies, for example, utilizes a thermometer for rating attitudes towards a particular elected official or candidate for office on a scale of 0 to 100, cold to hot [50]; such a valance measure of prior attitude toward the wildlife in the conflict is a critical moderating factor in both affective response and risk perception.

Understanding the principal mechanisms that exacerbate human-wildlife conflict is crucial to the development of successful policies aimed at diminishing or eradicating risks associated with the conflict, including public health risks. Our study contributes three innovations in understanding and predicting support for wildlife management policies that would be of interest to policy makers and activists seeking to redirect public discourse in human-wildlife conflict. First, we use the theoretical anchor of the Narrative Policy Framework (NPF) [51] to inform the structure of the narrative risk message treatments. The NPF posits that narratives are measurable across policy domains, because narratives themselves have a reliable structure that includes elements such as characters, setting, and plot [52]. As such, we can isolate narrative mechanisms (e.g., the casting and portrayal of characters) to more precisely and reliably test their effects. In tandem, through simulated social media posts, we test the power of image to intensify responses in comparison with the narrative text alone. Second, we stay true to risk theory by assessing the mediating role of affect, but include the valance of affect (positive to negative) in shaping the dual nature of impact associated with wildlife—benefits and costs. Third, we consider how prior attitudes toward wildlife moderate these effects, making an analytical step toward better integrating prior experiences beyond a simple covariate and identifying how risk messages are received across critical and distinct audiences.

We use a moderated multiple mediator model to test whether prior attitudes towards wildlife moderate the mechanisms (affect and risk perception) through which narrative risk communication influences public support for wildlife management strategies. Our model simultaneously estimates three stages of the causal path between narrative communication and public opinion: i) the effect of narrative and image treatments on affective response, ii) the effect of affective response on risk perception (perceived positive or negative impact and perceived likelihood of impact), and iii) the effect of risk perception on support for wildlife management proposals. The model also tests whether these effects vary based on prior attitudes towards wildlife. We examine these dynamics in the context of on-going conflict between humans and fruit bats, or flying foxes, in Australia. Fig 1 depicts our conceptual frame and analytical design.

With respect to the first stage, we anticipate that narrative casting wildlife as villains results in higher negative affective response (e.g., frustrated, upset, disgusted), while narrative casting wildlife as victims results in higher positive affective response (e.g., hopeful, inspired, determined). We expect the addition of images alongside these narratives will intensify affective response in the respective predicted directions. With respect to the second stage, we expect a more negative affective response to engender more negative perceived impacts of wildlife on humans, while a more positive affective response will engender more positive perceived

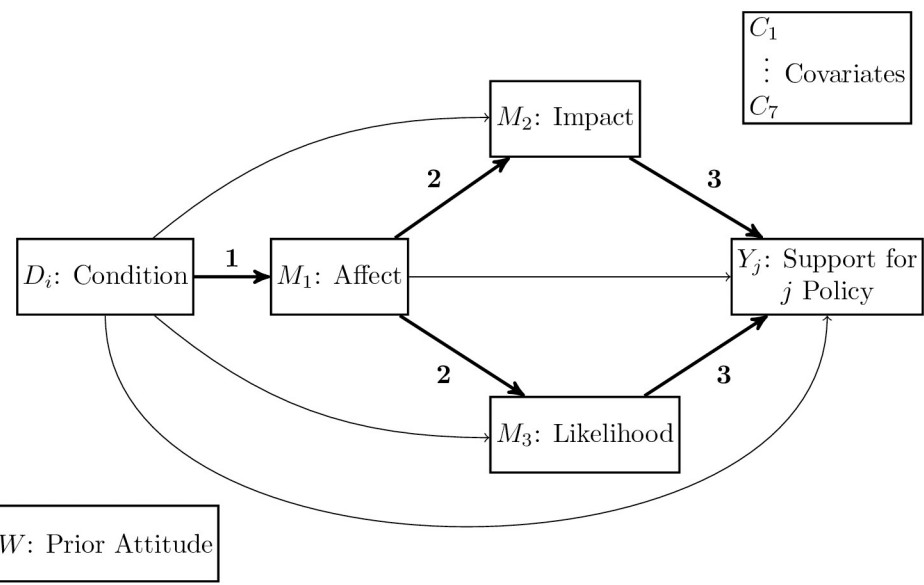

**Fig 1. Concept design of conditional process model.** Bold arrows depict the path of interest. Numbers indicate stage of mediation. Some arrows included in the model are excluded from the diagram for clarity: all paths depicted in the diagram are tested for moderation by prior attitude towards bats ($W$), and covariates are included in the prediction of all outcome variables ($M_1$, $M_2$, $M_3$, $Y_j$).

impacts of wildlife on humans. We predict that the intensity of the affective response, as opposed to the direction (positive or negative), will determine the perceived likelihood of impacts on humans. With respect to the third stage, we expect negative and likely perceived impacts of wildlife on humans to predict more support for wildlife relocation, and positive and likely perceived impacts of wildlife on humans to predict more support for wildlife protection. As for moderating effects, we expect prior attitudes towards wildlife to condition the strength of these relationships such that warm prior attitudes correspond with more favorable responses and perceptions towards wildlife, while cool prior attitudes correspond with less favorable responses and perceptions towards wildlife.

## Case description

We chose to examine whether and how narrative risk communication shifts public opinion in the context of human conflict and coexistence with flying foxes in urban and suburban areas of Australia. Flying foxes, also known as Old World fruit bats (genus *Pteropus*), serve as critical pollinators for Australia's forests [53]. In the populated states of Queensland and New South Wales, agricultural expansion and urban development has removed critical food sources and habitat for flying foxes [54]. As such, contact between humans and bats has increased, intensifying conflict [55, 56]. The loss of habitat has resulted in these animals exploiting resources in human-dominated environments, such as fruit in orchards and urban gardens, contributing to a new phenomenon of urban bats [54]. Aside from public health concerns (e.g., spread of infectious disease), impacts to quality of life (e.g., noise and smell) and local economy (e.g., destruction of commercial food crops, destruction of property) have fueled regular coverage in conventional and social media platforms. Similar strained relations between humans and flying foxes have been noted all over their range of distribution, contributing to individual and collective management decisions that threaten the longevity of these species (e.g., mass culling, destruction of roosts) [4, 6].

Urban flying foxes present an ideal case to model polarity in affective response and risk perception. Flying foxes exemplify the duality of wildlife in that they provide critical ecosystem services in the form of long distance pollination and cultural value, but are also a source of nuisance, loss of property, and infectious disease. Existing research highlights negative public perception towards bats [10, 57, 58], but also suggests a correlation between education and more positive attitudes towards bats [4, 59, 60]. Social media has offered an outlet for sharing information, pictures, and commentary related to bats that reflect and proliferate both negative and positive sentiments and appeals [59, 61]. Flying foxes are unlike microbats in that they are comparatively larger, and have furry fox-like faces [9]. Our anecdotal observations suggest that images of flying foxes engender either adoration and sympathy or vitriol and disgust. Non-profit groups and government agencies have invested in public outreach and education campaigns to shift public attitudes about bats via social media and other channels, while the efficacy of different messaging techniques and the mechanisms involved in changing public opinion in this context are still ripe for investigation [59] and serve as the motivation of this study.

## Research design

We fielded a survey experiment between May 2, 2019 and May 20, 2019 on a sample of 3,200 adult Australians in Queensland and New South Wales (see S1 Research design for survey methodology). This study received approval from the Montana State University Institutional Review Board, IRB approval number: ES111516. Consent to participate was obtained from study participants in written/electronic form.

We randomly assigned participants to one of six conditions: non-narrative without an image, non-narrative with an image, victim narrative without an image, victim narrative with an image, villain narrative without an image, and villain narrative with an image (see S1 Table for the number of respondents per condition). The non-narrative statement without an accompanying image was used as the baseline to determine relative effects of narratives and images.

Each condition was presented in the style of a Facebook post in which the identifying information about the fictitious author was censored (Fig 2 depicts the victim and villain narrative conditions with images; see S2 Fig for depictions of the non-narrative and non-image conditions). The conditions were intentionally written to be approximately the same length, sentence format, and grade school reading level. The villain narrative condition consists of a narrative casting bats as deleterious to humans. The victim narrative condition casts bats as suffering because of humans. Finally, unlike the narrative treatments, the non-narrative condition contains objective information and does not characterize bats as having positive or negative impacts on humans or the environment. The same image was used for all image treatments.

Prior to exposure to treatment, we asked respondents to rate their attitude towards bats on scale of 0 to 100, with lower numbers corresponding to a very cold or less favorable attitude towards bats and higher numbers corresponding to a very warm or more favorable attitude towards bats. The mean prior attitude towards bats was 52, which we interpret as neutral towards bats. We subsequently interpret a rating one standard deviation (26.67) below the mean with a cool attitude towards bats, and one standard deviation above the mean with a warm attitude towards bats.

To measure affect we presented survey participants with a series of emojis labeled with corresponding emotions and ask respondents to rate the intensity with which they felt each emotion upon reading the Facebook post (S3 Fig). We chose emojis/emotions based on emotions

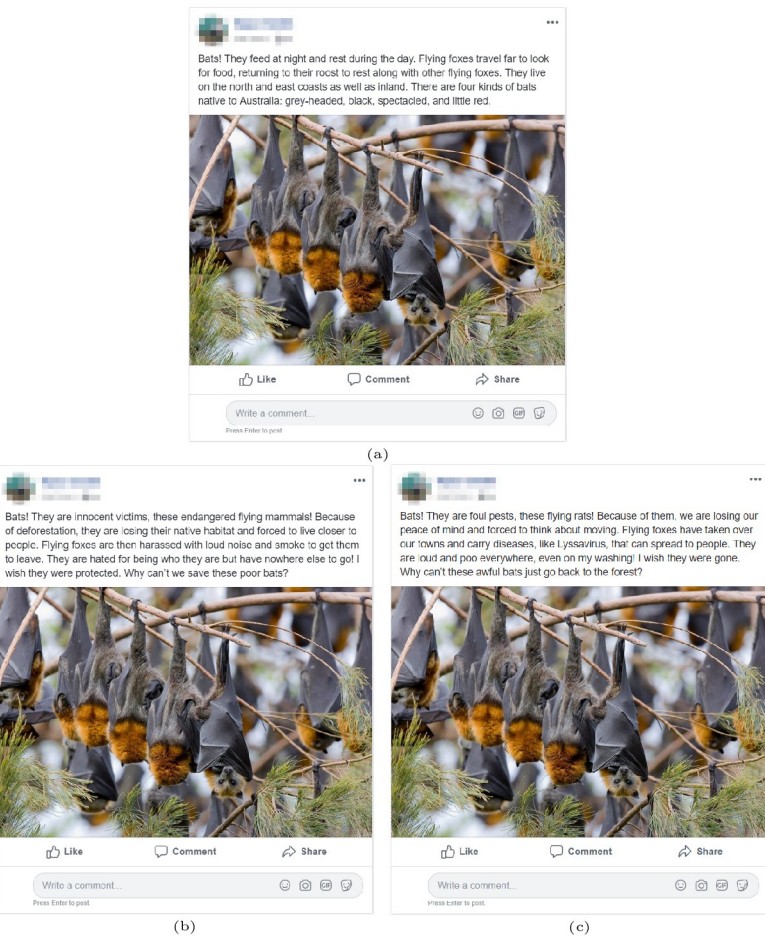

**Fig 2.** Depiction of non-narrative (a), victim narrative (b), and villain narrative (c) conditions, with image treatment. See S2 Fig for depictions of non-narrative, victim narrative, and villain narrative conditions without image treatment. Survey participants were randomly assigned only one of the six conditions. Photo used with permission from Bruce Thomson.

featured in the Positive and Negative Affect Schedule [62]. We used the intensity ratings to generate a single affective score ranging from positive to negative (S3 Table).

We then asked survey participants whether they perceived the impacts of bats to be positive or negative on three dimensions: i) economic, ii) quality of life, iii) health; respondents assessed these impacts at both the personal and community levels. We then asked them to rate their perception of the likelihood of each of these types of impacts occurring within the next year. We consolidated these measures into two additive indices, one for overall perceived impact and one for overall perceived likelihood of impacts occurring.

Finally, we measured respondents' level of support for five possible approaches to bat management. We characterize these approaches as prioritizing protection for bats in the form of state and federal regulations, prioritizing protection for humans in the form of dispersal of bats from urban and suburban roosts, and a compromise between protection for bats and protection for humans in the form of habitat restoration away from urban areas. (Additional details regarding operationalization and measurement, including covariates, can be found in S1 Research design; see S3 Table for summary statistics, and S4 Table for means and standard deviations of key variables across conditions).

We built a custom conditional process model to quantify relative direct and indirect effects using the PROCESS macro for SPSS [63]; a conceptual representation of the model is presented in Fig 1 (please refer to the S1 Research design for equations, and S4 Fig for a statistical diagram of the model). Our objective is to test the mediating effects of affect and risk perception and the moderating effects of prior attitudes towards bats in shaping the impact of narratives and images on support for bat management policies. Therefore, though we control for other indirect effects in our model, we focus on the indirect effects depicted in Fig 1, which consist of three stages. The first stage examines the relative effect of treatment conditions compared with the baseline non-narrative message without an image on affective response. The second stage of mediation examines the effect of affective response on two measures of risk perception: perception of positive or negative impact of bats on individuals and their community, and the likelihood that these impacts will occur in the next year. The final stage of mediation examines the effect of perceived impacts and perceived likelihood of impacts on support for bat management policies.

## Results

Overall, we find evidence to confirm our expectation that the effects of narrative and image on support for bat management policies operate indirectly through affect and perceived risk perception (see S5 Table for all relative indirect effect coefficients and bootstrapped standard errors). The villain narrative, in particular, has the highest impact on support for bat policies through affect and risk perception. We expected the addition of an image alongside narratives to intensify affective response, but found that this was only true some of the time and not always in the direction we anticipated. An image of flying foxes presented alongside a victim narrative intensified a negative affective response; but an image presented alongside a villain narrative dampened, or reduced, the negative affective response. This pattern is reflected in a comparison of the means across conditions (S4 Table) and in the comparative magnitudes of the indirect effects (S5 Table). Importantly, the addition of an image to the non-narrative message does not appear to influence support for bat management policies through affect and risk perception, suggesting that the combination of *narrative* and image is particularly important in shifting public opinion through affective response and risk perception.

As S5 Table demonstrates, affective response to narratives and the perceived *likelihood* of impacts are important mechanisms mediating the effect of victim and villain narratives on support for all bat management policies (federal and state protection, roost dispersal from residential and public spaces, and habitat restoration), while affective response and the perception of *positive or negative* impacts are important mediators in the effect of narratives on federal and state protection, and roost dispersal from public spaces. The perception of positive or negative impacts appears to have a weaker role than the perceived likelihood of impacts in mediating the effects of narrative on support for roost dispersal from residential areas and for habitat restoration.

The following sections describe each stage of the path of mediation in detail to provide more context for interpretation of the indirect effects, including where prior attitudes matter.

### First stage

The first stage of the conditional process model (Fig 1, path labeled "1") quantifies the relative effects of narrative and image treatments on affective response compared to the baseline non-narrative message without an image (S6 Table). A significant increase in R-squared between between a conditional versus unconditional first stage suggests these effects are moderated by prior feelings towards bats ($\Delta R^2 = 0.141$, $F(5, 3181) = 133.124$, $p = 0.001$). Therefore, we

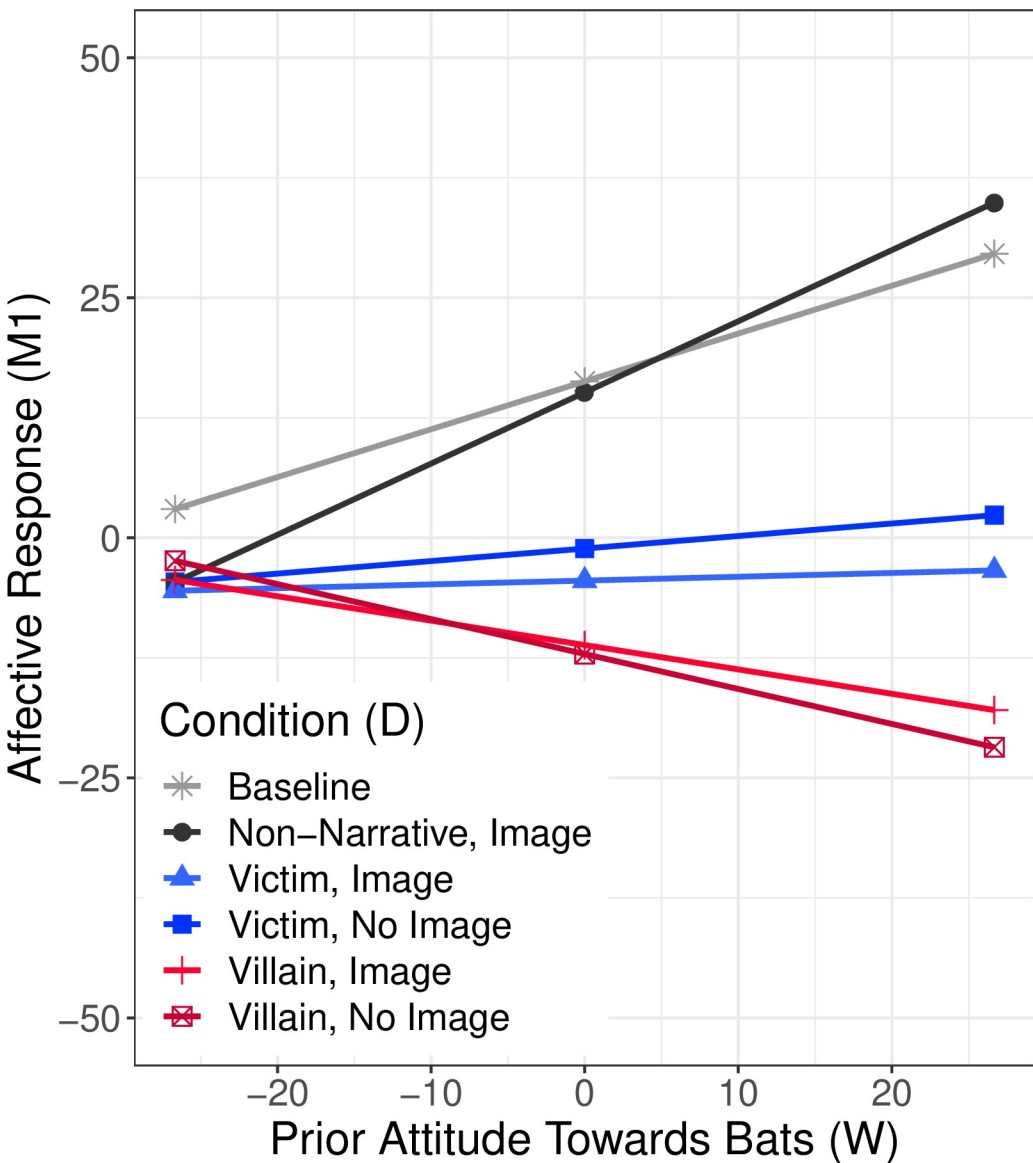

**Fig 3. Moderation of the effect of treatment conditions on affective response by prior feelings towards bats.**

present the estimated effects for participants at the mean (neutral towards bats), one standard deviation above the mean (warm attitudes towards bats), and one standard deviation below the mean (cool attitudes towards bats) (Fig 3).

Probing the interaction between treatment condition and affective response reveals the victim and villain narrative treatments prompt participants with neutral priors towards bats to react more negatively than participants with neutral priors receiving the baseline message. Reactions to the villain treatments are generally more negative than reactions to the victim treatments. The addition of images to the narratives does not substantively impact these effects.

With respect to participants with strong priors, we characterize narratives casting bats as villains to be challenging participants with warm priors towards bats and affirming participants with cool priors, while narratives casting bats as victims are characterized as affirming

participants with warm priors and challenging participants with cool priors. Among those with warm prior attitudes towards bats, exposure to the villain (challenging) narrative results in the most negative reaction. The addition of an image of bats appeared to slightly offset the effect of the challenging narrative (Fig 3). On the other hand, exposure to the victim (affirming) narrative results in a neutral affective response among participants with warm priors—in fact, a reflection of mixed positive and negative emotions (see S7 and S8 Tables). Among this population, the addition of an image of bats to the victim (affirming) narrative increases negative affective response, generating more negative emotions (e.g., sad) or fewer positive emotions (e.g., hopeful) than the victim narrative alone. Not surprisingly, the addition of an image to the non-narrative message results in a more positive affective response among this population compared to the non-narrative message without an image.

Among those with cool prior attitudes towards bats, both victim and villain narrative treatments result in more negative affective responses compared with responses to the baseline condition (Fig 3). The addition of an image to any message—narrative or non-narrative—results in a slightly more negative affective response among these participants, suggesting that images of bats intensify negative reactions among those with cool priors. On the whole, the effect sizes of narratives and images one standard deviation below neutral are much smaller than effect sizes one standard deviation above neutral, meaning that those with warm priors towards bats react most divergently to different types of risk communication while those with cool priors towards bats appear to react negatively to most risk communication with the exception of non-narrative message without an accompanying image.

## Second stage

Affective response to stimuli is theorized to shape risk perception; therefore, the second stage of the conditional process model quantifies the effect of affect on the two dimensions of risk perception, impact and likelihood (Fig 1, paths labeled "2").

We find a more positive affective response to treatment conditions induces more positive perceived impacts of bats on respondents' personal and community economics, quality of life, and health (S6 Table). Conversely, a more negative affective response to treatment condition corresponds to more negative perceived impacts of bats on the individual and the community. This effect is not moderated by prior feelings towards bats ($\Delta R^2 = 0.0001$, $F(1, 3179) = 0.612$, $p = 0.434$).

Affective response has an inverse relationship with perceived likelihood of impacts. In general, a more negative (less positive) affective response to risk messages corresponds to the perception that impacts are more likely to be felt in the next year, whereas a more positive affective response to risk messages corresponds to the perception that impacts are less likely to be felt in the next year. This effect is moderated by prior feelings towards bats ($\Delta R^2 = 0.001$, $F(1, 3179) = 5.370$, $p = 0.021$), and is stronger among those with cool prior attitudes towards bats (S6 Table, S5 Fig). In other words, those with cool priors towards bats are more sensitive to affect when evaluating the likelihood of risk than those with warm priors.

## Third stage

The third stage of the conditional process model quanitfies the effect of risk perception on support for bat management policies: i) protection for bats at the federal and state levels, ii) dispersal of bats from residential and public areas, and iii) habitat restoration aimed at attracting bats to less populated areas (Fig 1, paths labeled "3").

**Bat protection.** The perception of positive impacts from bats corresponds with more support for federal and state protection of bats (see S9 Table for coefficients corresponding to all

outcome variables). Conversely, the perception of negative impacts from bats corresponds to less support for bat protection. The effect on support for state protection is moderated by prior attitudes towards bats ($\Delta R^2$ = .001, $F(1, 3175)$ = 6.001, $p$ = .014, S6 Fig) and the effect is slightly larger for those participants with cool priors, meaning that support for state protection for bats among people who generally dislike bats is more dependent on whether they perceive positive or negative impacts of bats on themselves and their community.

The perception that impacts are more likely to occur in the next year corresponds with less support for federal and state protection of bats. We do not find evidence that this effect is moderated by prior attitudes towards bats ($\Delta R^2$ = .000, $F(1, 3175)$ = .501, $p$ = .479), meaning the effect size is similar for those with warm, neutral, or cool attitudes towards bats prior to treatment.

This result is consistent with the dual perception of hazards and benefits associated with wildlife in human-wildlife interaction. If hazards associated with bats are perceived to be more likely to occur, it is reasonable to be wary of regulations protecting bats; conversely, if hazards associated with bats are perceived to be less likely to occur, then support for bat protection is relatively harmless to the individual or the community. On the other hand, if benefits associated with bats are perceived to be less likely to occur, it is reasonable to be more supportive of regulations protecting bats in order to potentially reap those benefits in the more distant future. Conversely, if benefits associated with bats are perceived as more likely to occur, then additional protections for bats could be perceived as less urgent.

**Bat dispersal.** The perception of positive impacts from bats on individual participants and their community corresponds with less support for bat dispersal from residential areas and public spaces. Conversely, the perception of negative impacts from bats corresponds with more support for bat dispersal.

The perception that positive or negative impacts from bats are more likely to occur in the next year also corresponds with more support for bat dispersal. This effect is moderated by prior attitudes towards bats ($\Delta R^2$ = 0.002, $F(1, 3175)$ = 8.294, $p$ = 0.004 dispersal from residential areas; $\Delta R^2$ = 0.002, $F(1, 3175)$ = 7.331, $p$ = 0.007 dispersal for public spaces; S9 Table and S7 Fig). In general, the perception that impacts are more likely to occur in the next year corresponds to more support for bat dispersal. However, this effect is stronger among participants with warmer priors towards bats. This means the perceived likelihood of impacts associated with bats occurring in the short term is more important in determining whether those with warm priors towards bats are more or less supportive of bat dispersal, while those with cool priors towards bats are more likely than those with warm priors to support bat dispersal either way, and their support for bat dispersal is less dependent on their perception of the likelihood of impacts (see S7 Fig).

**Habitat restoration.** The effects of risk perception (impact and likelihood) on support for habitat restoration away from urban areas is similar to their effects on support for bat protection (see S9 Table). The perception that bats have more positive impacts on individuals and their community corresponds with more support for habitat restoration, and the more likely impacts are perceived to occur in the next year, the less support for habitat restoration.

Unlike support for bat protection, the effect of perceived likelihood on habitat restoration is moderated by prior attitudes towards bats ($\Delta R^2$ = 0.002, $F(1, 3175)$ = 5.881, $p$ = 0.015); the effect is stronger among those with warm priors and weaker among those with cool prior attitudes towards bats (S8 Fig). This reveals that support for habitat restoration among those who do not like bats prior to treatment is less sensitive to their perception of impact likelihood.

It is worth noting that the magnitude of the effects of the risk perception measures on habitat restoration are lower than those for bat protection. This is not so surprising given that habitat restoration is the compromise approach to mitigating human-bat conflict, aiming to

protect both bats and humans. A closer examination of the means across experimental conditions reveals that habitat restoration garners consistently higher support than state or federal level bat protection across all groups (S4 Table). Though the directions of the effects are the same between risk perception and support for bat protection and risk perception and support for habitat restoration, the lower magnitudes of the effects of perceived impact and likelihood on habitat restoration may simply reflect less variation in support for habitat restoration and therefore less explanatory power. Thus, this result is consistent with the characterization of habitat restoration away from urban areas as a potential opportunity for consensus.

## Discussion

Our findings reveal that narrative risk communication in the context of human-wildlife conflict is more effective than non-narrative scientific communication in mobilizing support for different approaches to conflict mitigation. We demonstrate that the effectiveness of narrative risk communication in influencing public opinion is a product of changes in affective response and risk perception. Thus, our results support theory-based expectations that persuasiveness is dependent on the emotional transportation of narratives and images in risk messaging [25, 31]. Specifically, narratives casting wildlife as victims or as villains induce more negative affective responses than non-narrative messaging, and these effects are more pronounced among those with warmer prior attitudes towards wildlife. Those with warm priors react positively to non-narrative messaging, and comparatively react much more negatively to narratives casting wildlife as victims or villains.

We find evidence to support our expectation that images intensify negative affective responses in most cases, with the exception of affective responses from people with warm prior attitudes towards bats. Those with warm priors reacted more positively to an image with a non-narrative message, and their negative reaction to the villain narrative appeared marginally softened by the presence of an image of bats.

Our results support our expectation that affect drives risk perception. Negative affect contributes to the perception of more negative impacts of wildlife on individuals and their community, regardless of prior attitudes towards wildlife. We find that negative affect also drives the perception that impacts are more likely to occur in the short term, and this effect is stronger among those with cool prior attitudes towards wildlife.

Finally, the effects of risk perception on support for human-wildlife conflict management approaches also conform to our expectations. The perception of more negative impacts of wildlife on humans leads to less support for wildlife protection, more support for wildlife relocation, and less support for habitat restoration, while the perception that impacts are more likely to occur also leads to less support for wildlife protection, more support for wildlife relocation, and less support for habitat restoration. Moreover, we find evidence that some of these effects are conditional on prior attitudes towards wildlife, uncovering potential opportunities for more effective targeted messaging.

Our results show that people with warmer prior attitudes towards wildlife demonstrate more variation in affective response to risk communication, reacting more positively to non-narrative messaging, reacting with mixed emotions to the victim narrative, and reacting negatively to the villain narrative. Negative affect contributes to greater risk perception among this population, and their support for some management approaches is linked to their perception of whether or not impacts from wildlife are likely to occur in the short term. Notably, support for wildlife relocation is strongly linked to perceived likelihood of impacts among those with warm priors towards wildlife, but their support for wildlife protection and habitat restoration is less sensitive to perceived likelihood of impacts. To summarize, those who like wildlife are

inclined to support wildlife protection and habitat restoration regardless of risk perception, but can be moved to support wildlife relocation from urban areas if perceived impacts of wildlife are likely to occur.

People with cooler prior attitudes towards wildlife demonstrate more negative affective response to narrative risk communication in comparison with non-narrative communication. This negative affective response leads to more negative and more likely perceived impacts associated with wildlife. Among this population, support for state protection of wildlife is more dependent on perception of positive/negative impacts than it is for those with warm priors towards wildlife. However, this population's position on wildlife relocation and habitat restoration away from urban areas is *not* dependent on perceived likelihood of impacts. To summarize, those who dislike wildlife are inclined to support wildlife relocation and oppose habitat restoration regardless of risk perception. However, this population can be moved in the direction of supporting wildlife protection if primed to perceive more positive impacts of wildlife. Future research should attempt to target this population and explore the effectiveness of mechanisms priming positive impacts specifically, including casting wildlife as a hero in a narrative risk message.

It is worth noting that survey participants' prior attitudes towards bats had the largest effect in the first stage, affective response to treatment conditions, compared to subsequent stages on the path of mediation. We found the moderating effect of prior attitudes towards bats to be comparatively smaller in predicting variation in risk perception and support for policies. This suggests that prior attitudes influence initial receptivity and reactivity to certain types of messaging more than influencing the way affective response translates to perceived risk, and the way perceived risk translates to support for policies.

Our experimental design features treatments intended to mimic narrative communication style observed in the current social media environment, and test the effects of this type of communication on affective response, risk perception, and public opinion. The connection between narrative risk communication, affective response, risk perception, and support for wildlife management policies may inform messaging strategies for campaigns aimed at communicating the risks and benefits associated with human interaction with wildlife, particularly using social media platforms. Our findings that indirect effects of narratives and images on support for wildlife management policies are moderated by prior attitudes towards wildlife highlight the importance of matching narrative messages and images to audience, which may be useful for targeted messaging in the social media environment.

However, since the effect of narrative on public opinion works through affective response and risk perception, these conditional indirect effects also reveal a pathway for how some people might be particularly vulnerable to disinformation regarding risks to humans or wildlife in human-wildlife interaction. While our work may provide insight for communicators in the sciences and across stakeholder groups, we urge careful consideration of the ethical questions raised in the practice of narrative risk communication, including a thorough examination of the goals (e.g., persuasion, comprehension) and attention to the level of accuracy maintained [31]. Science communicators may want to pay special attention to issues of trust and credibility, and future work ought to evaluate the impacts of narrative risk communication on messenger credibility. Reasonable hypotheses offer competing predictions: narrative communication may increase trust in communicators through perceived authenticity and accessibility, or decrease trust from perceived intention to manipulate [31, 34].

While this study explores the mechanisms through which narrative risk communication influences public opinion about wildlife management in the context of human-wildlife conflict, there is room for broadening scope conditions. For example, this study tested the effect of adding the same image across all narrative messages on support for policies through affect and

risk perception. Future work may test the effect of adding images depicting wildlife in different contexts alongside narrative messages. Furthermore, this study was limited to narratives casting wildlife as victims or villains; future studies ought to explore whether casting wildlife as heroes in narrative risk communication elicits more positive affective response, works to accentuate positive impacts of wildlife, and maximizes support for longer-term wildlife management, particularly when targeted towards populations with cool prior attitudes towards wildlife.

## Supporting information

**S1 Research design.**
(PDF)

**S1 Results.**
(PDF)

**S1 Fig. Map of Queensland and New South Wales.** Blue represents postcodes without known bat roosts; pink represents postcodes with known bat roosts.
(TIF)

**S2 Fig.** Depiction of non-narrative (a), victim narrative (b), and villain narrative (c) conditions, without image treatment.
(TIF)

**S3 Fig. Emojis representing Positive and Negative Affect Schedule emotions comprising affective response measure.**
(TIF)

**S4 Fig. Statistical diagram of conditional process model.**
(TIF)

**S5 Fig. Conditional effects of affective response on risk perception.**
(TIF)

**S6 Fig. Conditional effects of risk perception on support for state-level protection.**
(TIF)

**S7 Fig. Conditional effects of risk perception on support for dispersal of bats from urban roosts.**
(TIF)

**S8 Fig. Conditional effects of risk perception on support for habitat restoration.**
(TIF)

**S1 Table. Number of respondents per treatment condition.**
(TIF)

**S2 Table. Block randomization and sample distribution.**
(TIF)

**S3 Table. Sample summary statistics.**
(TIF)

**S4 Table. Means and standard deviations across experimental conditions.**
(TIF)

**S5 Table. Relative indirect effects on outcome variables.**
(TIF)

**S6 Table. Relative effects of treatment conditions on mediating variables.**
(TIF)

**S7 Table. Means and standard deviations for survey participants with warm prior attitudes towards bats.**
(TIF)

**S8 Table. Means and standard deviations for survey participants with cool prior attitudes towards bats.**
(TIF)

**S9 Table. Relative direct effects of conditions, mediators, and moderator on outcomes of interest.**
(TIF)

## Acknowledgments

The authors would like to thank Peggy Eby for providing bat roost location data, Nita Bharti, Taylor Carlson, Alison Peel, and Raina Plowright for feedback on survey design, and two referees and the editor for their excellent comments and suggestions.

## Author Contributions

**Conceptualization:** Sara K. Guenther, Elizabeth A. Shanahan.

**Data curation:** Sara K. Guenther, Elizabeth A. Shanahan.

**Formal analysis:** Sara K. Guenther, Elizabeth A. Shanahan.

**Funding acquisition:** Sara K. Guenther, Elizabeth A. Shanahan.

**Investigation:** Sara K. Guenther, Elizabeth A. Shanahan.

**Methodology:** Sara K. Guenther, Elizabeth A. Shanahan.

**Project administration:** Sara K. Guenther, Elizabeth A. Shanahan.

**Resources:** Sara K. Guenther, Elizabeth A. Shanahan.

**Software:** Sara K. Guenther, Elizabeth A. Shanahan.

**Supervision:** Sara K. Guenther, Elizabeth A. Shanahan.

**Validation:** Sara K. Guenther, Elizabeth A. Shanahan.

**Visualization:** Sara K. Guenther, Elizabeth A. Shanahan.

**Writing – original draft:** Sara K. Guenther, Elizabeth A. Shanahan.

**Writing – review & editing:** Sara K. Guenther, Elizabeth A. Shanahan.

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
