## [Decision Letter · Decision Letter 0]

12 Nov 2020

PONE-D-20-27944

Communicating risk in human-wildlife interactions: how stories and images move minds

PLOS ONE

Dear Dr. Guenther,

Thank you for submitting your manuscript to PLOS ONE. After careful consideration, we feel that it has merit but does not fully meet PLOS ONE’s publication criteria as it currently stands. Therefore, we invite you to submit a revised version of the manuscript that addresses the points raised during the review process.

We look forward to receiving your revised manuscript.

Kind regards,

Christian Vincenot, Ph.D.

Academic Editor

PLOS ONE

Journal Requirements:

Additional Editor Comments:

I agree with the two reviewers in their assessment of the value of this interesting study. Please see their comments, especially the ones regarding the method.

I would be grateful if you could also address the following points:

1. Personally, I very much welcome your work and ideas to improve communication strategies in conservation. I feel that we have not achieved the greatest success possible because conservation communication has been too rigid. In a piece with Vincent Florens, we had called for "broader conservation strategies" including "marketing persuasion principles" to fight the ongoing culls of the Mauritian flying fox Pteropus niger in response to conflicts with farmers (see https://science.sciencemag.org/content/362/6413/409.1). Hence, I am receptive to the point that made here (i.e. that narratives should be used for conservation communication), yet my concern is that it cuts both ways. If we start advocating for "narrative" messages in favor of bats, the scientific community (and to a larger extent the conservation community) will not be legitimate anymore in arguing against the same methods being used by adversaries. Currently, we are still in a position in which we can oppose the latter by stating that what we offer to the public is evidence-based knowledge, devoid of political aims, subjective feelings or "narratives". As somebody who has communicated extensively in the media about bat-farmer conflicts (esp. regarding Mauritius), I reckon that this stance is precious to convince the public. Deviating from this line and starting to promote and spread narratives might open a Pandora box. This really needs to be somehow acknowledged in your manuscript I think. I will not request for a solution to this complex problem be detailed (this could be the object of another self-standing paper), but I feel that a bit of discussion is needed to inform the readership about the risks. Furthermore, I reckon that you have been very careful in your wording and recommendations. But as your message will be (mis)interpreted, I think that you might want to prevent your recommendation to be seen as a call for the use of propaganda-like communication in conservation. In case my point is not clear, please feel free to get back to me by e-mail (vincenot@i.kyoto-u.ac.jp).

2. The case that you studied here (FFs in Australia) is very interesting and valuable. You might want to show that it is illustrative of a global issue, as flying foxes are persecuted and threatened all over their range. See https://science.sciencemag.org/content/355/6332/1368 and references therein.

3. Similarly, there were few studies on perception of flying foxes, which adds to the relevance and importance of your work. You may want to stress this by citing the few existing works (and perhaps discuss some of their results compared to what you observe here). See for instance:

https://link.springer.com/article/10.1007%2Fs10745-017-9905-6

https://www.sciencedirect.com/science/article/pii/S2351989415000190

4. Minor, but "mega bat" should be spelled "megabat". (Please note that, although still used in common language, it is actually considered now better replaced by "Old World fruit bat'", or more strictly, Yinpterochiroptera/Pteropodid, in scientific publications.)

Reviewers' comments:

Reviewer's Responses to Questions

**Comments to the Author**

1. Is the manuscript technically sound, and do the data support the conclusions?

Reviewer #1: Yes

Reviewer #2: Yes

2. Has the statistical analysis been performed appropriately and rigorously? 

Reviewer #1: Yes

Reviewer #2: Yes

3. Have the authors made all data underlying the findings in their manuscript fully available?

Reviewer #1: Yes

Reviewer #2: Yes

4. Is the manuscript presented in an intelligible fashion and written in standard English?

Reviewer #1: Yes

Reviewer #2: Yes

5. Review Comments to the Author

Reviewer #1: Intro – previous attitudes and mechanism of change – please explictly speak about congruent or incongruent change

Final para – it is not clear why bats were chosen, please write short para about public perception of bats and finish the para with argument(s) that bats are perfect examples of human-wildlife conflict.

Aside from public health 104

concerns (e.g., spread of infectious disease) – see e.g., Musila, S., Prokop, P., & Gichuki, N. (2018). Knowledge and perceptions of, and attitudes to, bats by people living around Arabuko-Sokoke Forest, Malindi-Kenya. Anthrozoös, 31(2), 247-262.

L. 267 Conversely, the perception of negative impacts from bats corresponds to less

support for bat protection. – negative perception of animals = low conservation support, see Gunnthorsdottir, A., 2001. Physical attractiveness of an animal species as a decision factor for its preservation. Anthrozoös 14, 204–215

Prokop, P., & Fančovičová, J. (2017). Animals in dangerous postures enhance learning, but decrease willingness to protect animals. Eurasia Journal of Mathematics, Science and Technology Education, 13(9), 6069–6077.

A figure with visual presentation of villain and victim treatment is required

Reviewer #2: This is a nice study. The theoretical contributions do not stand out and I am not convinced there are any, but the practical implications are clear and valuable. If nothing else, this is a nice demonstration of how to use narratives and imagery to create positive change in the context of HWCC issues. And I think that is enough to warrant publication. I think you could draw clearer theoretical implications by exploring (in the literature review and discussion) the theoretical mechanisms of the effects, but I do not see that as essential. Finally, the manuscript is well written, with ideas emerging and flowing intuitively and consistently. Despite a complex analysis, you do a good job presenting and interpreting the key findings. See below for specific comments.

ABSTRACT

Perhaps there’s another way to phase “human-flying fox conflict,” which may easily refer to a conflict about foxes that fly humans. I know foxes flying humans is nonsense, but there’s still ambiguity about the phrase. I’m not requiring or even requesting a change, but you might consider it. You might also highlight in the abstract that flying foxes are a type of bat. Of course they are, but some readers might not know that. I, for one, love flying foxes. Their babies are super cute and even some adults, too.

INTRO

You have a parenthetical aside, “severity/cost * likelihood,” which creates ambiguity over the use of the forward slash. I presume you mean it as “severity-slash-cost” and not “severity divided by cost.” But I had to think about it for a moment. Perhaps you can use a multiplication symbol (×) rather than an asterisk (because asterisks go with slashes as mathematical notation) or omit “cost” altogether.

You cited Green and Brock to raise the concept of transportation. I wish to call your attention to Dahlstrom (2014; https://doi.org/10.1073/pnas.1320645111), who did a nice job discussing the influence of narratives in the contexts of science and environmental communication. I am not sure your citations 25 through 28 (narratives in risk contexts) included comparisons of narrative and non-narrative texts. If they do not, you might cite another one of Dahlstrom’s works, which made that comparison in the context of climate change denial (see Dahlstrom and Rosenthal, 2018; https://doi.org/10.1177/1075547018766556).

There has also been recent work in the environmental communication arena looking at the linkages between positive and negative affect and benefit and risk perception (e.g., Kahlor et al., 2019; https://doi.org/10.1080/17524032.2019.1699136). That work seems germane to the point you make after mentioning transportation.

You should clarify early on that you are portraying bats, not humans, as villains. This was unclear until the research design. You also need to clarify the source of positive and negative impacts. When I initially read your series of predictions, I inferred a counter-intuitive mediation model in which the portrayal of victims leads to positive emotion, which leads to positive perceived impacts (on the victims). I came to understand you meant the impact of the victims on humans, but only after reviewing S1.

Please clarify how prior attitudes will condition the effects of narratives and images. You clearly indicated a positive moderation effect of images on the effect of narratives. What kind of effect do you anticipate for attitudes and how does that reflect extant scientific knowledge?

In my summary comment, I stated that the theoretical contributions are lacking. I wish to give you a brief example. On page 4, you write, “Understanding the principal mechanisms that exacerbate human-wildlife conflict is crucial to the development of successful policies….” Then you identify the “Narrative Policy Framework” as a theoretical anchor. However, you do not explain any of the mechanisms of that framework. In fact, that is the only instance of “Narrative Policy Framework” in your manuscript (excluding the references).

DESIGN

The two paragraphs about flying foxes seem to be about the context of your study, which is not clearly about the research design. Perhaps you could put them in a dedicated “context” section. I think that would go well at the end of the literature review.

Based on your Figure 1 note, I presume all direct and indirect paths were moderated by W. You should add “W” in parentheses as you do with M1, M2, M3 and Yj, and locate it immediately after “prior attitude towards bats.” I think that will help the reader understand your implementation of PROCESS.

RESULTS

Your first statement implies overall support for your predictions about narratives and images. You had predicted “the use of an image with narratives will intensify affective response.” Table S4 does not appear to show that. It appears after adding the image, the victim condition resulted in a more negative affective score and the villain condition resulted in a more positive affective score. I suspect those differences are significant, which would run counter to your prediction, no? I would like to see more discussion of the treatment effects on affective score, which would help make sense of the subsequent effects. I understand the treatment effects are conditioned on priors, but your prediction in the literature review first indicates the unconditioned treatment effects before regarding the conditioning by priors.

Your analysis involves, essentially, a multiple-moderated mediation of a 2 x 3 factorial treatment effect. This is a complex analysis, not from a technical standpoint, but from how best to present it to readers. I think you have done generally a good job focusing the analysis on a few key findings. However, I wonder if there is a more structured organizational approach. For example, when you state (at the end of the literature review) the findings you expect to see, you might structure it in the three-stage format of your results.

DISCUSSION

Several of the observed effects are very small (R^2 < .01). I hope you can add a note in the discussion about the practical significance of those findings.

You invoke the concept of transportation in explaining the effect of narrative. However, I am not convinced transportation was the mechanism. Why not something simpler like involvement? Do you believe the brief narratives contained enough story to create a transporting effect? If so, why? And then what does the addition of images mean for your theoretical account? Green describes transportation as an “integrative melding of attention, imagery, and feelings.” What happens when the narrative provides the images for the reader?

6. PLOS authors have the option to publish the peer review history of their article (what does this mean?). If published, this will include your full peer review and any attached files.

Reviewer #1: No

Reviewer #2: **Yes: **Sonny Rosenthal

---

## [Author Response · Author response to Decision Letter 0]

7 Dec 2020

Thank you for your excellent feedback and engagement with our paper. We are certain the paper is stronger with the revisions inspired by your comments. Please see Response Memo for responses to specific comments.

---

## [Editor Report · Decision Letter 1]

10 Dec 2020

Communicating risk in human-wildlife interactions: how stories and images move minds

PONE-D-20-27944R1

Dear Dr. Guenther,

We’re pleased to inform you that your manuscript has been judged scientifically suitable for publication and will be formally accepted for publication once it meets all outstanding technical requirements.

Kind regards,

Christian Vincenot, Ph.D.

Academic Editor

PLOS ONE

Additional Editor Comments (optional):

Thank you for taking into considerations all the comments and addressing them. Congratulations on the interesting study.
---

## [Editor Report · Acceptance letter]

16 Dec 2020

PONE-D-20-27944R1 

Communicating risk in human-wildlife interactions: how stories and images move minds 

Dear Dr. Guenther:

I'm pleased to inform you that your manuscript has been deemed suitable for publication in PLOS ONE. Congratulations! Your manuscript is now with our production department. 

Kind regards, 

on behalf of

Dr. Christian Vincenot 

Academic Editor

PLOS ONE